# Deep Learning-Based Muscle Segmentation and Quantification of Full-Leg Plain Radiograph for Sarcopenia Screening in Patients Undergoing Total Knee Arthroplasty

**DOI:** 10.3390/jcm11133612

**Published:** 2022-06-22

**Authors:** Doohyun Hwang, Sungho Ahn, Yong-Beom Park, Seong Hwan Kim, Hyuk-Soo Han, Myung Chul Lee, Du Hyun Ro

**Affiliations:** 1Department of Orthopedic Surgery, Seoul National University College of Medicine, Seoul 03080, Korea; dhhwang@snu.ac.kr (D.H.); tjdgh914758@snu.ac.kr (S.A.); oshawks7@snu.ac.kr (H.-S.H.); leemc@snu.ac.kr (M.C.L.); 2Department of Orthopedic Surgery, Seoul National University Hospital, Seoul 03080, Korea; 3Department of Orthopedic Surgery, Chung-Ang University Hospital, Seoul 06973, Korea; whybe1122@gmail.com (Y.-B.P.); ksh170177@nate.com (S.H.K.); 4Connecteve Co., Ltd., Seoul 03080, Korea

**Keywords:** sarcopenia, deep learning, screening, plain radiograph, segmentation, total knee arthroplasty, osteoarthritis

## Abstract

Sarcopenia, an age-related loss of skeletal muscle mass and function, is correlated with adverse outcomes after some surgeries. Here, we present a deep-learning-based model for automatic muscle segmentation and quantification of full-leg plain radiographs. We illustrated the potential of the model to predict sarcopenia in patients undergoing total knee arthroplasty (TKA). A U-Net-based deep learning model for automatic muscle segmentation was developed, trained and validated on the plain radiographs of 227 healthy volunteers. The radiographs of 403 patients scheduled for primary TKA were reviewed to test the developed model and explore its potential to predict sarcopenia. The proposed deep learning model achieved mean IoU values of 0.959 (95% CI 0.959–0.960) and 0.926 (95% CI 0.920–0.931) in the training set and test set, respectively. The fivefold AUC value of the sarcopenia classification model was 0.988 (95% CI 0.986–0.989). Of seven key predictors included in the model, the predicted muscle volume (PMV) was the most important of these features in the decision process. In the preoperative clinical setting, wherein laboratory tests and radiographic imaging are available, the proposed deep-learning-based model can be used to screen for sarcopenia in patients with knee osteoarthritis undergoing TKA with high sarcopenia screening performance.

## 1. Introduction

Total knee arthroplasty (TKA) is an effective treatment for end-stage knee osteoarthritis [1,2]. Despite its benefits for relieving pain and restoring mobility, TKA is characterised by several postoperative complications, including the need for blood transfusion, acute kidney injury, prosthetic joint infection and periprosthetic tibiofemoral fractures with varying incidence rates [3,4,5,6]. To prevent such adverse outcomes, risk factors for postoperative complications of TKA have been explored [7,8,9].

Sarcopenia, characterised as age-related loss of skeletal muscle mass and function, has attracted a great deal of interest because of its reported association with an increased likelihood of poor clinical outcomes, including falls, fractures, physical disability and mortality [10]. Recent studies have showed that sarcopenia independently predicts adverse outcomes of various surgical procedures, including TKA [11,12,13,14]. Therefore, there is increasing recognition of the importance of preoperative risk stratification by screening orthopaedic surgery patients for sarcopenia [15,16,17,18].

Several tools are available for sarcopenia screening. Calf circumference (CC) is an inexpensive and readily available screening method. There is accumulating evidence that it correlates with both muscle mass and sarcopenia [19,20,21]. However, as the European Working Group on Sarcopenia in Older People (EWGSOP2) does not recommend CC as a screening tool for sarcopenia due to age-related changes in fat deposits and variable skin elasticity [10], its performance has mainly been studied in Asian populations [20,22,23]. The EWGSOP2 and Asian Working Group for Sarcopenia (AWGS) recommended the SARC-F questionnaire as a screening tool in primary healthcare settings [10,24]. However, several studies have reported low-to-moderate sensitivity of this tool to detect sarcopenia, such that a substantial number of potential patients likely go unrecognised [25]. There have been numerous efforts to boost its sensitivity for screening purposes by lowering the cutoffs, adding extra items and combining it with other examinations, but there is as yet no consensus regarding the best tool for screening sarcopenia [26].

As both the EWGSOP2 and AWGS suggest muscle quantity or mass as the confirmatory diagnostic criterion for sarcopenia, a radiological assessment of muscle quantity may represent an alternative method for screening sarcopenia, directly reflecting muscle loss in patients. A radiological assessment by magnetic resonance imaging (MRI) and computed tomography (CT) is considered the gold standard for the noninvasive assessment of muscle mass due to the high accuracy of these modalities. The abdominal skeletal muscle area at the third lumbar level of a single cross-sectional CT image is reported to have correlation with the whole-body skeletal muscle mass, and thus is being deployed to assess sarcopenia [27]. However, the need for time-consuming manual segmentation and potential interobserver variability are major barriers to clinical application [28,29]. In addition, imaging modalities such as CT and MRI are not always available for orthopaedic surgeons in the preoperative setting, hindering their application for sarcopenia screening. Therefore, there is a need for an alternative and more accessible tool to assess muscle quantity in the field of orthopaedics [30].

Here, we propose a novel method of screening for sarcopenia in patients undergoing orthopaedic surgery involving the lower extremities in the preoperative setting. We adopted a convolutional neural network (CNN)-based model for automatic muscle segmentation on full-leg weight-bearing plain radiographs. CNNs are the current state-of-the-art artificial intelligence technique for medical image classification and segmentation. The ability to screen for sarcopenia on full-leg plain radiographs prior to surgical procedures will aid orthopaedic surgeons in terms of risk stratification and patient selection.

The main objective of this study was to present a deep-learning-based muscle segmentation and quantification model using full-leg plain radiographs and illustrate its potential to predict sarcopenia in patients undergoing TKA, which was then confirmed by a bioelectrical impedance analysis (BIA).

## 2. Materials and Methods

### 2.1. Study Subjects

This retrospective single-centre study was conducted on two separate cohorts after obtaining institutional review board approval (IRB no. H-2009-181-1161).

For training and validation of the model, healthy volunteers with no history of trauma or prior orthopaedic surgery, enrolled between January 2011 and November 2012, were reviewed (cohort A). Cohort A consisted of two subgroups: a young group consisting of 128 young adults aged 19–35 years (79 females, 61.7%) and an older group consisting of 99 patients aged 60–69 years (51 females, 51.5%). A total of 227 full-leg lower extremity plain radiographs were used as the training set. The muscle segmentation model presented in this study represents a fully automated deep learning system, which was developed, trained and validated using cohort A data.

Patients scheduled for primary TKA to treat degenerative knee arthritis, enrolled between May 2018 and April 2021, were reviewed to test the developed model and explore its potential for predicting sarcopenia (cohort B). Subjects with adequate preoperative BIA, determined using the InBody S10 device (InBody Co. Ltd., Seoul, Korea), were included in cohort B. A total of 633 patients were initially enrolled. We excluded patients who underwent simultaneous bilateral TKA (*n* = 5), or had inadequate tissue hydration (extracellular water (ECW) ratio > 0.4; *n =* 175), no available preoperative plain radiographs (within 8 weeks before scheduled surgery; *n* = 4), low-quality full-leg plain radiographs (*n* = 45) or severe obesity (body mass index (BMI) > 35 kg/m^2^; *n =* 1) [31,32,33]. A total of 403 patients (54 males, 349 females) were included in the test set. Data for cohort B were used to evaluate the performance of the developed muscle segmentation model and estimate the muscle quantity of the patients. The machine-learning-based sarcopenia prediction model presented in this study was developed, trained and validated using cohort B data.

### 2.2. Data Acquisition

All data were collected from the electronic medical records of our institution. Patient baseline characteristics, laboratory data and BIA data were collected. Sarcopenia was determined according to the cutoffs for appendicular skeletal muscle index (SMI) suggested by the AWGS 2019 (<7.0 kg/m^2^ for males, <5.7 kg/m^2^ for females) [33]. SMI is defined as the height-adjusted appendicular skeletal muscle mass and was calculated automatically by the InBody S10 device (InBody Co. Ltd., Seoul, Korea). The baseline characteristics, laboratory data and SMI were compared between the sarcopenic and normal groups (Appendix A).

Full-leg weight-bearing plain radiographs were acquired using a consistent scanning protocol. The subjects stood barefoot with the feet together in the “stand at attention” position, with the patellae oriented forward. The acquired radiographic images were collected using the picture archiving and communication system (PACS) of our institution.

One of the authors (SA) trained in the segmentation task prepared 227 manually annotated images using Adobe Photoshop (Adobe, Mountain View, CA, USA) using a standardised protocol, to ensure consistency of the data. Muscles of the bilateral thighs and calves were annotated. The upper border of the thigh was determined as the greater trochanter of the femur (lateral) and gluteal fold (medial), and the lower border of the thigh was determined as the lateral epicondyle of the femur (lateral) and medical epicondyle of the femur (medial). The visible parts of the soleus and gastrocnemius muscle on coronal plane radiographs were annotated as the bilateral calf regions. Each ground truth segmentation was reviewed and revised by an orthopaedic surgeon with 14 years of experience (DR) (Figure 1).

### 2.3. Model

An overview of the study is shown in Figure 2.

#### 2.3.1. Data Preprocessing

All full-leg plain radiographs and ground truth masks were cropped and resized to 240 × 1200 pixels. Pixel values of grayscale images ranging from 0 to 255 on plain radiographs were normalised (Z-score normalisation) so that each image had a mean pixel value of 0 and standard deviation of 1 [34].

#### 2.3.2. Model Architecture

U-Net shows high performance in medical image segmentation, attributed to its skip connections that allow feature extraction without a significant loss of resolution [35]. In this study, we adopted a U-Net-like architecture to develop an optimal model to segment muscle from full-leg plain radiographs. To increase its feature extraction capability, we added the squeeze-and-excitation (SE) block to the encoding path of the network, which adaptively recalibrated channel-wise feature responses to significantly improve CNN performance, without marked increases in model complexity or computational burden [36]. In addition, we applied group normalisation instead of batch normalisation, which is the most commonly used normalisation method. Retaining the resolution of the original image was important for segmenting muscle from plain radiographs but had the potential to adversely affect model performance (as a larger image size will inevitably result in a smaller batch size due to the limited memory). Instead, robust normalisation methods such as group normalisation, which performs well even with a small batch size, can be used [37].

#### 2.3.3. Network Training

The Adam optimiser was used to train our model [38]. The Lovász–Hinge loss was used as a loss function and backpropagation was applied to the network weights for training [39]. The training batch size was set to 4. We used a global learning decay strategy that reduced the learning rate by 90% when the loss reached a plateau. The training and model performance assessment were conducted using a 5-fold cross-validation to address potential overfitting due to the small dataset. Data augmentation strategies included rotation (−10, +10), horizontal flip and scaling (0.9, 1.1).

#### 2.3.4. Model Performance Evaluation

The most widely adopted metrics for evaluating models for semantic segmentation are the mean intersection-over-union (IoU) value and dice similarity coefficient (DSC) [40,41]. The performance of the developed model was evaluated using both the training (cohort A) and test sets (cohort B) (Appendix A).

#### 2.3.5. Muscle Volume Estimation

A total of 403 test images from cohort B were fed into the developed model for muscle segmentation based on full-leg plain radiographs, to estimate the muscle volume. From the output segmentation mask, the predicted muscle area (*A*) was estimated as the sum of nonzero pixels multiplied by pixel size. We assumed that a single leg was a cylinder to estimate the predicted muscle volume (PMV) as follows: *PMV* = *πa* (*d_L_* + *d_R_*)/4. Here, *d_L_* and *d_R_* are the longest diameters of the mid-thigh region of the left and right extremities, respectively. The mid-thigh muscle area has been reported to be a good predictor of whole-body skeletal muscle mass, as it is highly sensitive to changes therein [10]. The mid-thigh diameter was measured automatically by postprocessing in our model. A Pearson’s correlation analysis was performed between the PMV and SMI.

#### 2.3.6. Machine Learning Model for Sarcopenia Prediction

Thirty preoperative variables were initially chosen as candidate predictors based on previous studies [12,14,42,43,44]. The preoperative variables included patient demographic information, comorbidities, laboratory data and the proposed sarcopenia marker (PMV) (Appendix A). The model for sarcopenia prediction was developed, trained and validated based on PMV and the baseline characteristics of patients of cohort B.

For prediction of sarcopenia, the binary classification (sarcopenia/normal) model of XG Boost was used. The XGBoost model is one of the most commonly used machine learning models for solving both regression and classification problems and has been widely adopted to classify and predict medical events [3,45,46]. The synthetic minority oversampling technique (SMOTE) was utilised to overcome the potential bias arising from class imbalances, by creating synthetic minority class samples (the sarcopenia group in this case) [47,48]. Key features for the XGBoost classification were selected. The comparison of key features between sarcopenia and normal group was conducted. A stratified 5-fold cross-validation of the training dataset was performed to obtain the optimal degree of model complexity. Training accuracy was evaluated from the mean area under curve (AUC) value of the receiver operating characteristic (ROC). Along with the AUC and model accuracy, the sensitivity and specificity of the binary classification were also investigated. To understand the decision-making process, the importance of each feature in the machine learning model was also determined by calculating a “gain”, which refers to the relative contribution of the corresponding feature to the model by taking each feature’s contribution for each tree in the model [49].

### 2.4. Statistical Analysis

Statistical analyses were performed using RStudio for Windows (ver. 1.2.5033; RStudio, Boston, MA, USA). Nominal data are shown as percentages and were analysed by two-sided Pearson’s χ^2^ test or Fisher’s exact test. Continuous data are shown as the mean ± SD and were analysed using Student’s *t* test. In all analyses, *p* < 0.05 was taken to indicate statistical significance.

## 3. Results

The proportion of patients who had undergone primary TKA and had sarcopenia was 8.5% (34/369). There were significant differences between the sarcopenia and nonsarcopenia groups in age, BMI, preoperative total protein and haemoglobin (Hb) levels, skeletal muscle index (SMI) and PMV (Table 1). The patients with sarcopenia were older than those without the condition (74.6 ± 6.5 vs. 70.5 ± 6.5 years, *p* = 0.001), and had a lower BMI (23.9 ± 3.4 vs. 26.7 ± 3.2 kg/m^2^, *p* < 0.001), preoperative total protein level (6.7 ± 0.4 vs. 7.1 ± 0.4 mg/dL, *p* < 0.001), preoperative Hb (12.3 ± 1.2 vs. 13.1 ± 1.8 g/dL, *p* = 0.008), SMI (5.5 ± 0.6 vs. 7.4 ± 1.1 kg/m^2^, *p* < 0.001) and PMV (6972.4 ± 1354.6 vs. 8418.4 ± 1634.8 cm^3^, *p* < 0.001).

The deep learning model developed for the muscle segmentation of full-leg plain radiographs achieved a mean DSC of 0.944 (95% CI 0.936–0.951) and mean IoU value of 0.959 (95% CI 0.959–0.960) for the developmental training cohort A. For the test set images of cohort B, it achieved a mean DSC of 0.913 (95% CI 0.910–0.916) and mean IoU value of 0.926 (95% CI 0.920–0.931).

Pearson’s correlation analysis was performed between PMV and SMI, and the correlation coefficient was 0.654 (*p* < 0.001) (Figure 3).

The stratified fivefold AUC value of the XGBoost model after internal validation with the test set (cohort B) was 0.988 (95% CI 0.986–0.989) (Figure 4). The classification model had an accuracy of 0.945 (95% CI 0.941–0.950), sensitivity of 0.970 (95% CI 0.962–0.978) and specificity of 0.926 (95% CI 0.920–0.931).

Of the 30 preoperative variables, 7 key predictors were selected for the model: age, BMI, total protein, albumin, Hb, bilirubin and PMV. Feature importance ranks, which indicate the relative importance of input features, were calculated to understand the decision-making process of the XGBoost model (Figure 5). PMV was the most important feature in the decision-making process (feature importance: 0.179). Along with the PMV, BMI (0.164), bilirubin (0.158), preoperative Hb (0.132), albumin (0.131), total protein (0.123) and age (0.113) were key features for sarcopenia prediction; PMV, BMI, Hb, total protein and age also showed statistical significance (*p* < 0.05) in a univariate analysis.

## 4. Discussion

The most important findings of this study were that our CNN-based deep learning model showed high performance in terms of automatic muscle segmentation of full-leg plain radiographs, while the XGBoost classification model, which included several important patient features such as PMV, showed high performance for predicting sarcopenia.

The feature importance ranks of our sarcopenia prediction model showed that PMV can serve as a feasible sarcopenia marker. The feature importance score of PMV suggested that it is the most important factor for predicting sarcopenia. Along with PMV, it is notable that several serum markers also contributed to the decision-making process. Total bilirubin has been reported to be positively correlated with SMI, although it was not significantly different between sarcopenia and normal groups [42]. Bilirubin is one of the most active endogenous antioxidant molecules and is therefore thought to have a protective effect against the progression of sarcopenia. Albumin and total protein are considered good markers of nutritional status and were reported to be low in patients with sarcopenia [43,44]. A low Hb level was also reported to be associated with sarcopenia, although the pathophysiology of this relation has not been explored [50]. The selection of serum markers that are correlated with sarcopenia indicates that the decision-making process of the XGBoost classifier was reasonable.

To our knowledge, among all reported sarcopenia screening models, ours showed the highest performance (AUC 0.988, 95% CI 0.986–0.989) [25,33,51,52]. We have developed a pipeline involving both deep-learning- and machine-learning-based models, to take both radiographic images and patient baseline characteristics into consideration (which can be easily obtained in the preoperative clinical setting). Moreover, there have been no previous reports of automatic muscle segmentation or quantification of full-leg plain radiographs for sarcopenia screening. There are studies reporting the potential of chest radiographs to analyse body composition, but not for the purpose of assessing sarcopenia [53]. Although surgeons can use other imaging modalities, such as CT, to investigate the risk of sarcopenia, we evaluated muscle mass on full-leg plain radiographs as they are routinely obtained in TKA patients and are therefore appropriate for screening purposes. Our method could also be applied to other surgical procedures in which full-leg plain radiographs are obtained as part of routine examinations, which would be particularly useful in the field of orthopaedics. Our proposed sarcopenia screening method may not be applicable in a community healthcare setting, where neither laboratory testing nor radiographic imaging is available. In the preoperative setting, on the other hand, sarcopenia can be screened with near-perfect sensitivity and specificity using only elementary laboratory data and full-leg plain radiographic images.

This study had several limitations. Firstly, there were some barriers to a reasonable estimation of the muscle volume on full-leg plain radiographs. Although our deep-learning-based model was generally successful in performing muscle segmentation on full-leg plain radiographs, it tended to slightly overestimate the number of pixels in the muscle layer. Moreover, the model had difficulty in segmenting the medial and the upper margin of the thigh muscle region (gluteal fold).

The small dataset size and homogeneity of the training set may have affected the performance and robustness of the model and given rise to the above issues. Specifically, the difference in patient distribution between the training set (healthy volunteers with no history of osteoarthritis or trauma) and test set (patients with end-stage osteoarthritis undergoing TKA) may have decreased model performance. In addition, several images did not have a distinct gluteal fold and were therefore labelled with reference to the opposite side during the annotation process, to create the ground truth segmentation mask. Such annotation inconsistency may have contributed to the ambiguity in the training process. Therefore, extensive training of the model with additional full-leg plain radiographs from a wide array of patients is required to improve the accuracy of the model.

Secondly, due to fatty infiltration of skeletal muscle in patients with sarcopenic obesity, the estimated muscle mass of patients may not always represent the actual muscle mass [54]. Features such as radio-opacity of the plain radiographs, which may reflect fatty infiltration, will be incorporated in future studies.

For the classification model of sarcopenia, an external validation was not performed, which raises questions regarding its validity when applied to other cohorts. In addition, the predominance of females in our study population may have adversely affected the model feature selection process. As the study population consisted only of patients undergoing TKA, the key features and performance of the model may not generalise to preoperative settings for other surgical procedures. Further studies are required to expand the number of available training sets; the proposed model should also be applied to independent test datasets from other institutions to verify its validity.

As the diagnostic criteria involve not only muscle mass, functional tests of muscle strength and physical performance, such as gait speed, muscle grip strength, the get-up-and-go test and peak expiratory flow, should also be included in the sarcopenia screening process [33]. We expect that incorporation of such tests in future studies will render our approach to screen sarcopenia using full-leg plain radiographs more legitimate.

In a population of 90,438 patients who had undergone primary TKA, Ardeljan et al., reported that 16.7% had sarcopenia. The patients with sarcopenia had longer hospital stays and increased odds of falls, lower extremity fracture, reoperation, 2-year implant-related complication rates, higher surgery costs and higher rates of postoperative blood transfusion and complications within 90 days [17]. As it is an indicator of the risk of adverse events during the postoperative period, cases considered at high-risk of sarcopenia could be confirmed by dual-energy X-ray absorptiometry (DEXA) if necessary, which is the gold standard for quantifying muscle mass [10]. As a modifiable risk factor [55,56], patients suspected of having sarcopenia based on screening can also be managed prior to TKA to improve clinical outcomes. After TKA, they should be treated with more caution and attend more frequent follow-up visits.

## 5. Conclusions

Here, we presented a novel method for screening sarcopenia in the preoperative clinical setting, using a fully automated deep learning model to automatically segment and quantify the muscle layer on full-leg plain radiographs, and validated its potential to predict sarcopenia in patients undergoing TKA. In the preoperative clinical setting, wherein laboratory tests and radiographic imaging are available, our model showed high sarcopenia screening performance.

## Figures and Tables

**Figure 1 jcm-11-03612-f001:**
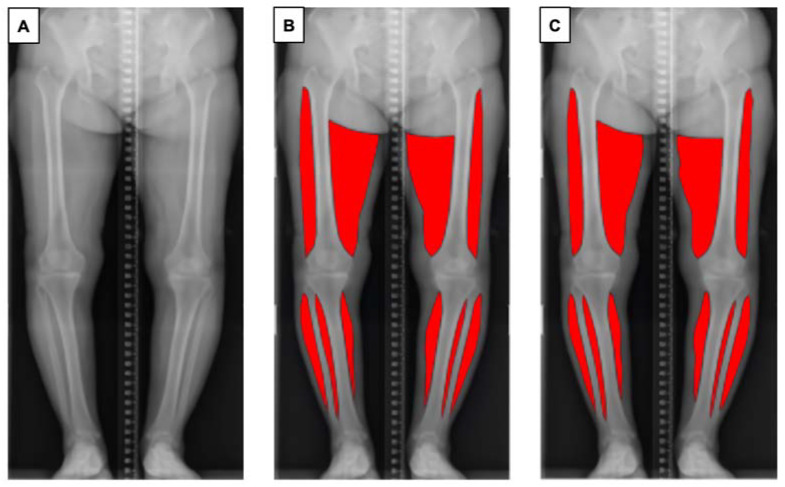
Overview of the pipeline for automatic muscle segmentation of full-leg plain radiographs. (**A**) Original radiographic image. (**B**) Ground truth mask of segmented muscle generated by the authors. (**C**) Segmented muscle predicted by the proposed model.

**Figure 2 jcm-11-03612-f002:**
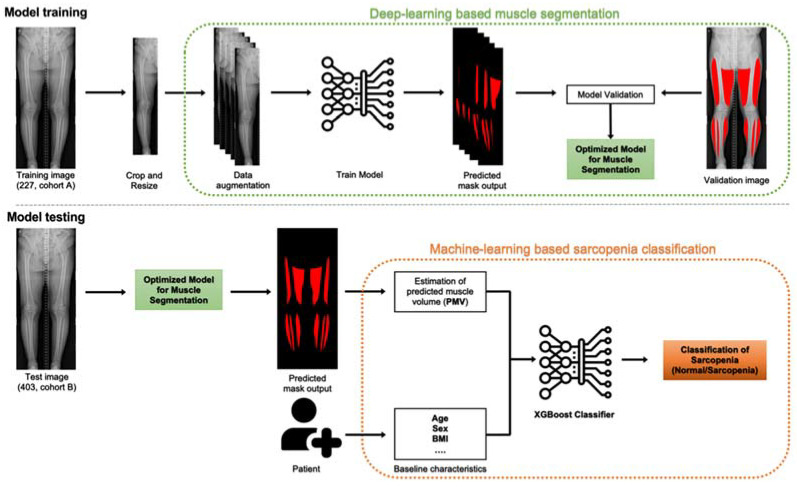
Model overview. Model training, left to right: the model is trained on input images, which undergo preprocessing (crop and resize) and augmentation (enriching the training set) before being fed into the convolutional neural network. The model is trained and validated by 5-fold cross validation. Hyperparameter tuning is conducted to optimise the model for muscle segmentation. Model testing, left to right: the optimised eXtreme Gradient Boosting (XGBoost) classification model is used to estimate predicted muscle volume (PMV) of the patients and, along with baseline characteristics, to classify patients into sarcopenia and normal groups.

**Figure 3 jcm-11-03612-f003:**
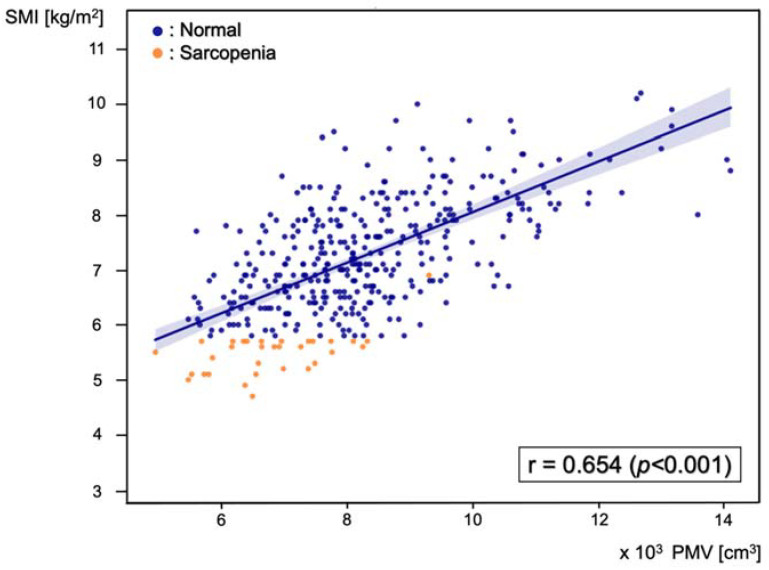
Scatter plot of the correlation between predicted muscle volume (PMV) and skeletal muscle index (SMI), with a Pearson’s correlation coefficient of 0.654 (*p* < 0.001). Blue dots represent patients in the normal group and orange dots represent patients in the sarcopenia group.

**Figure 4 jcm-11-03612-f004:**
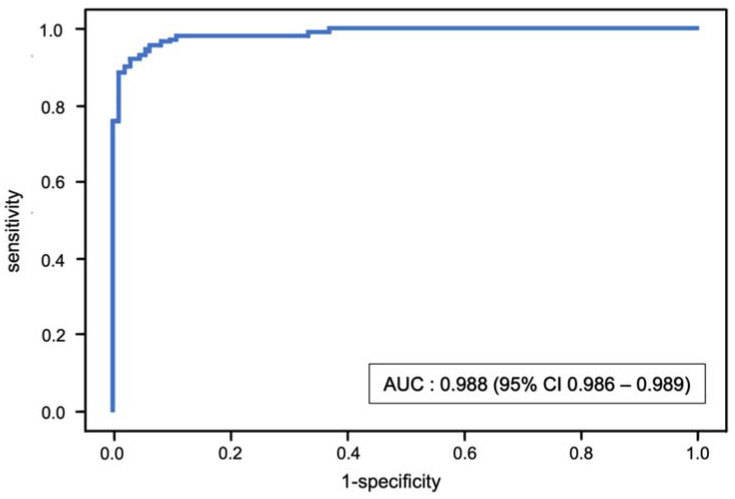
Receiver operating characteristic (ROC) curve and mean area under the curve (AUC) for the binary classification of sarcopenia using the XGBoost model.

**Figure 5 jcm-11-03612-f005:**
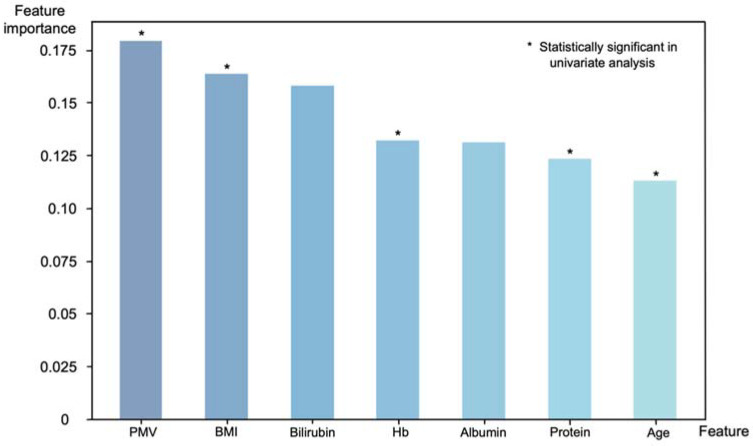
Feature importance with respect to the binary classification of sarcopenia using the XGBoost model. Features that showed statistical significance in a univariate analysis also ranked high in importance in the XGBoost model, with PMV being the most important features.

**Table 1 jcm-11-03612-t001:** Key features comparison of the sarcopenia and nonsarcopenia groups.

	Total Population (*n =* 403)		
	Sarcopenia		
Characteristics	Yes (*n =* 34)	No (*n =* 369)	*p*-Value
Sex (%)			
Female	32 (94.1)	319 (86.4)	0.266
Male	2 (5.9)	50 (13.6)	
Age (SD)	74.6 (6.5)	70.5 (6.5)	<0.001
BMI, kg/m^2^ (SD)	23.9 (3.4)	26.7 (3.2)	<0.001
Total Protein, mg/dL (SD)	6.7 (0.4)	7.1 (0.4)	<0.001
Albumin, g/dL (SD)	4.1 (0.3)	4.2 (0.4)	0.194
Hemoglobin, g/dL (SD)	12.3 (1.2)	13.1 (1.8)	0.004
Total Bilirubin, mg/dL (SD)	0.6 (0.3)	0.6 (0.2)	0.946
SMI, kg/m^2^ (SD)	5.5 (0.6)	7.4 (1.1)	<0.001
PMV, cm^3^ (SD)	6972.4 (1354.6)	8418.4 (1634.8)	<0.001

Values are shown as the mean ± standard deviation or number (%). Statistical significance was set at *p* < 0.05. BMI, body mass index. SMI, skeletal muscle index. PMV, predictive muscle volume.

## Data Availability

Not applicable.

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
