# Peer review of "Deep Learning-Based Muscle Segmentation and Quantification of Full-Leg Plain Radiograph for Sarcopenia Screening in Patients Undergoing Total Knee Arthroplasty"

_jcm, 2022, doi:10.3390/jcm11133612_

Round 1
Reviewer 1 Report
This paper presented an automatic method for sarcopenia screening in patients undergoing total knee arthroplasty (TKA). The method consists of two components, including 1) a deep learning model (U-Net) for muscle segmentation using full-leg radiograph followed by 2) a conventional machine learning model (XGBoost) for sarcopenia classification using estimated muscle volume along with baseline characteristics. Two groups of radiograph data are involved in this study. One group (cohort A) containing 227 radiographs of healthy volunteers is used for the training and evaluation of the deep learning-based segmentation model. Another group (cohort B) containing 403 TKA patients is used for the tuning and evaluation of the machine learning-based classification model. The experimental results showed that the proposed method can effectively segment the muscle from the full-leg radiographs of either healthy volunteers or TKA patients. The predicted muscle volume (PMV) works as the most important feature along with other key features (such as BMI, bilirubin, and preoperative Hb) in the prediction of sarcopenia in TKA patients.
Overall, this paper is well-written and easy to follow. The target problem (i.e., the sarcopenia screening in TKA patients) is of great clinical value and the author proposed to solve this problem in a novel way (i.e., using a deep learning model to estimate PMV and help the decision-making in sarcopenia screening).
I just have one major comment regarding the classification model used for the sarcopenia screening. The author proposed to use a deep learning model (U-Net) to perform the muscle segmentation but a conventional machine learning model (XGBoost) to perform the following sarcopenia prediction. I would like to know whether it is possible to achieve the sarcopenia prediction by using some kind of deep learning-based classification model, such as ResNet [R1]? If so, the author is suggested to conduct an experiment to investigate the model performance when using a deep learning-based classification model. If not, the author should give a discussion on this point to explain why this solution is infeasible or improper for their problem.
[R1] He, Kaiming, et al. "Deep residual learning for image recognition." Proceedings of the IEEE conference on computer vision and pattern recognition. 2016.
I also listed my minor comments below, which should be carefully addressed before I can suggest the acceptance of this paper.
Minor comments:
1) Line 25: Please specify the full name of “PMV” when it appears the first time in the abstract.
2) Line 159: “normalisation” -> “normalization”
3) Line 161, “as a larger image size will inevitably result in a smaller batch size”: This statement is a bit too strong. It would be better to add some conditions after that, for example, “as a larger image size will inevitably result in a smaller batch size due to the limited memory.”
4) Line 165, “Adam optimization”: Please cite the original literature of the Adam optimizer.
5) Line 165, “Lovasz-Hinge loss”: The author is suggested to explain more about the Lovasz-Hinge loss used for the segmentation network training. Why did the author choose this loss function? As far as I know, the U-Net for medical image segmentation is often trained by the Dice loss or the Cross-entropy loss or their combination. Did the author try to use this kind of loss function?
6) Line 168, “After the training has been completed, the performance of the model was assessed using 5-fold cross-validation …”: I am confused about the setting of the 5-fold cross-validation here. As my understanding, in a 5-fold cross-validation setting, the model should be trained from scratch for five rounds. During each round of training, four out of five folds of data samples are used for training. And the rest one fold of data samples are reserved for validation. But the statement here seems like the model is only trained for one time and validated for five times? The author is suggested to clarify the detailed steps of this 5-fold cross-validation in this study.
7) Line 182: The equation “PMV = pi*A/4 (dL + dR)” should be “PMV = pi*A*(dL + dR)/4”
8) Line 184: According to the statement, the mid-thigh muscle is a good predictor of whole-body skeletal muscle mass. But the predicted muscle volume (PMV = pi*A*(dL + dR)/4) is calculated based on the predicted muscle area (A), which includes not only the thigh muscle but also the calve muscle. Does that affect the following analysis in this study?
9) Line 193, “The model for sarcopenia prediction was developed, trained and validated based on radiographic images and … of cohort B.”: I think this statement is inaccurate. The model for sarcopenia prediction takes as input the predicted muscle mask rather than the radiographic image (see the bottom part of Figure 2).
10) Line 240: According to the author’s statement, there are 30 preoperative variables ranked by the feature importance. And the top-ranked 7 variables are selected as the model input. What’s the specific value of the threshold used in this selection? And how did the author determine this threshold?
11) Line 241: The author is suggested to provide more details about the calculation of the feature importance. Is there any formulation explicitly expressing how to calculate the feature importance?
Reviewer 2 Report
1) Introduction
a. The introduction clearly highlights the relevance of the topic: sarcopenia can severely affect the outcome of different surgical procedure and medical therapies. Furthermore, the chosen mean of assesment, knee xray, is widely available in the clinical context.
b. Line 44
i. I believe some relevant references must be added to this segment, namely this article demonstrating a very strong effect on outcomes of sarcopenia in transplant: https://pubmed.ncbi.nlm.nih.gov/35454842/
c. Line 64
i. I believe some relevant references must be added to this segment, namely this seminal paper on CT-based sarcopenia segmentation: https://pubmed.ncbi.nlm.nih.gov/26392166/
2) Materials and methods
a. The study cohort isreally large, more than adequate for the design.
b. The study design is appropriate.
c. Line 145: more details needs to be added regarding the normalization procedure. In general, the authors deployed advanced computational instruments in the study (ex. XGBoost), which require dedicated paragraphs or a relevant reference.
3) Results
a. Results are presented clearly, however there are some minor limitation to the exposition:
i. For clarity, a report of the correlation between the detected muscle volume and the BIA results could be useful to explore the method.
ii. If possible, report the total (gain/loss) function
4) Discussion
a. A relevant paper should be discussed in this paragraph, as it explored manual segmentation of Xray for Body composition analises: https://pubmed.ncbi.nlm.nih.gov/33551289/
b. Limitation of the study are expressed in details.
Round 2
Reviewer 1 Report
Thanks for the authors’ efforts in addressing my comments, which have been well-addressed in the revised version. I have no more comments but a recommendation of acceptance on this paper.